# Pathophysiology of Mild Hypercortisolism: From the Bench to the Bedside

**DOI:** 10.3390/ijms23020673

**Published:** 2022-01-08

**Authors:** Vittoria Favero, Arianna Cremaschi, Chiara Parazzoli, Alberto Falchetti, Agostino Gaudio, Luigi Gennari, Alfredo Scillitani, Fabio Vescini, Valentina Morelli, Carmen Aresta, Iacopo Chiodini

**Affiliations:** 1Department of Medical Biotechnology and Translational Medicine, University of Milan, 20133 Milan, Italy; vittoria.favero@unimi.it (V.F.); arianna.cremaschi@unimi.it (A.C.); chiara.parazzoli@unimi.it (C.P.); iacopo.chiodini@unimi.it (I.C.); 2Department of Endocrine and Metabolic Diseases, IRCCS, Istituto Auxologico Italiano, 20149 Milan, Italy; a.falchetti@auxologico.it; 3Department of Clinical and Experimental Medicine, University of Catania, 95123 Catania, Italy; agostino.gaudio@unict.it; 4Department of Medicine, Surgery and Neurosciences, University of Siena, 53100 Siena, Italy; luigi.gennari@unisi.it; 5Unit of Endocrinology and Diabetology “Casa Sollievo Della Sofferenza” Hospital, IRCCS, 71013 San Giovanni Rotondo (FG), Italy; alfredo.scillitani@gmail.com; 6Endocrinology and Metabolism Unit, University-Hospital S. M. Misericordia of Udine, 33100 Udine, Italy; vescini.fabio@aoud.sanita.fvg.it; 7Unit of Endocrinology, Fondazione IRCCS Cà Granda-Ospedale Maggiore Policlinico, 20122 Milan, Italy; valentina.morelli@policlinico.mi.it

**Keywords:** hypercortisolism, hypertension, diabetes, bone fragility

## Abstract

Mild hypercortisolism is defined as biochemical evidence of abnormal cortisol secretion without the classical detectable manifestations of overt Cushing’s syndrome and, above all, lacking catabolic characteristics such as central muscle weakness, adipose tissue redistribution, skin fragility and unusual infections. Mild hypercortisolism is frequently discovered in patients with adrenal incidentalomas, with a prevalence ranging between 5 and 50%. This high variability is mainly due to the different criteria used for defining this condition. This subtle cortisol excess has also been described in patients with incidentally discovered pituitary tumors with an estimated prevalence of 5%. To date, the mechanisms responsible for the pathogenesis of mild hypercortisolism of pituitary origin are still not well clarified. At variance, recent advances have been made in understanding the genetic background of bilateral and unilateral adrenal adenomas causing mild hypercortisolism. Some recent data suggest that the clinical effects of glucocorticoid (GC) exposure on peripheral tissues are determined not only by the amount of the adrenal GC production but also by the peripheral GC metabolism and by the GC sensitivity. Indeed, in subjects with normal cortisol secretion, the combined estimate of cortisol secretion, cortisone-to-cortisol peripheral activation by the 11 beta-hydroxysteroid dehydrogenase enzyme and GC receptor sensitizing variants have been suggested to be associated with the presence of hypertension, diabetes and bone fragility, which are three well-known consequences of hypercortisolism. This review focuses on the pathophysiologic mechanism underlying both the different sources of mild hypercortisolism and their clinical consequences (bone fragility, arterial hypertension, subclinical atherosclerosis, cardiovascular remodeling, dyslipidemia, glucose metabolism impairment, visceral adiposity, infections, muscle damage, mood disorders and coagulation).

## 1. Introduction

Mild hypercortisolism (mHC) is defined as biochemical evidence of abnormal cortisol secretion without the classical detectable manifestations of overt Cushing’s syndrome and, above all, lacking catabolic characteristics such as central muscle weakness, adipose tissue redistribution, skin fragility and unusual infections [1,2,3]. This condition has been called “subclinical Cushing’s syndrome” or “subclinical hypercortisolism” [4].

However, the term “subclinical” has been criticized because, although the mHC progression toward CS is very rare, mHC has been associated with hypertension, alterations in glucose and lipid metabolism, increased cardiovascular risk, bone fragility and mortality [5]. Given that the presence of mHC is frequently hidden, the term “hidden hypercortisolism” (HidHyCo) has been proposed for defining the condition of this asymptomatic and only biochemically evident form of hypercortisolism [6,7]. The European Society of Endocrinology utilizes the term “autonomous cortisol secretion” for defining the presence of mHC in patients with incidentally discovered adrenal masses (adrenal incidentalomas, AI) during imaging procedure for unrelated disorders [8].

The overall prevalence of mHC is estimated to be between 0.2 and 2% in individuals older than 60 years of age. This estimate derives from data showing that AI are present in 3% of 50-year-old patients (up to 10% in the elderly) [1,5] and that the prevalence of mHC in AI ranges between 5 and 30%. This high variability is due to the different diagnostic criteria used for defining mHC [1,3,4]. The presence of mHC is even more frequent (about 22–42%) in patients with bilateral AI [1,9] and with larger adrenal tumors (i.e., above 2.4 cm) [3]. Interestingly, however, the prevalence of the associated comorbidities is comparable between these two conditions [10]. Besides patients with AI, mHC was suggested to be more prevalent than expected in some populations at risk, such as patients with scarcely controlled or complicated hypertension and/or diabetes and patients with unexplainable bone fragility [6,7]. The prevalence of this condition in patients with incidentally discovered pituitary tumors has been poorly investigated. However, some data showed a 5% prevalence of pituitary mHC [11].

The biochemical criteria for diagnosing mHC are still a matter of debate, and, importantly, most data derive from studies on adrenal mHC in patients with AI. Many guidelines recommend the use of the 1 mg overnight dexamethasone suppression test (DST), but the cutoff to be used is still debated, with, for example, the European guidelines suggesting cortisol levels after DST between 1.9 and 5.0 μg/dL or >5.0 μg/dL as diagnostics for either “possible autonomous cortisol secretion” or “confirmed autonomous cortisol secretion”, respectively [8], while other experts suggest the use of DST with a cutoff of 1.8 μg/dL [4]. Basal morning plasma adrenocorticotroph hormone (ACTH), 24-h urinary free cortisol and/or late-night salivary cortisol may be used as additional tests for diagnosing adrenal mHC [4]. Finally, in the presence of mHC, the finding of ACTH levels above 20 pg/mL (or 30 pg/mL after corticotroph-releasing hormone stimulation) are suggestive of a pituitary origin [12].

Some recent data suggest that the clinical effects of GC (GC) exposure on peripheral tissues are determined not only by the amount of the adrenal GC production but also by the peripheral GC metabolism [13,14,15] and by the GC sensitivity [16] (ClinVar: https://www.ncbi.nlm.nih.gov/clinvar/?term=glucocorticoid+receptor. accessed on 30 December 2021). Interestingly, in subjects with normal cortisol secretion, the combined estimate of cortisol secretion, cortisone-to-cortisol peripheral activation by the 11 beta-hydroxysteroid dehydrogenase (11BHSD) enzymes and GC receptor (GR) sensitizing variants have been suggested to be associated with the presence of hypertension, diabetes and bone fragility, which are three well-known consequences of hypercortisolism [17,18].

Finally, a hypercortisolemic status may also characterize many medical disorders in which the hypothalamus–pituitary–adrenal axis is activated, such as in alcoholism, renal failure, poorly controlled diabetes and severe neuropsychiatric disorders [19]. To date, whether or not this functional hypercortisolism (named also pseudo-Cushing’s syndrome) may cause the same clinical consequences of a “neoplastic” hypercortisolism is under debate [20]. The pathogenesis and clinical implications of functional hypercortisolism will not be discussed in this review.

## 2. Material and Methods

We searched the following databases, including PubMed, Web of Science, SCOPUS and Google Scholar, using the following MeSH terms and/or their combinations: Cushing, Cushing’s disease, Cushing’s syndrome, hypercortisolism, mild hypercortisolism, subclinical hypercortisolism, subclinical Cushing’s syndrome, glucocorticoid, glucocorticoid receptor, pituitary adenoma secretion, ACTH, stress, adrenal incidentalomas, glucocorticoid-induced osteoporosis, bone fragility and cortisol, muscle and cortisol, cardiovascular risk and cortisol, mood disorders and cortisol, infection and cortisol, diabetes and cortisol, dyslipidemia and cortisol.

## 3. Pathophysiology of Pituitary and Adrenal Mild Hypercortisolism

### 3.1. Physiology of Cortisol Secretion and Metabolism

Cortisol production is regulated by the hypothalamus–pituitary–adrenal axis. Circadian input and stress stimuli act on the hypothalamus, leading to the release of hypothalamic neuroendocrine hormones, namely, corticotrophin-releasing hormone (CRH) and vasopressin (AVP), which stimulate the secretion of the adrenocorticotroph hormone (ACTH) from the anterior pituitary. In turn, ACTH acts in the fasciculata and in the reticularis zones of the adrenal cortex by binding to the melanocortin 2 receptor (MC2R), a G-protein coupled receptor, therefore activating the production of cyclic adenosine monophosphate by the adenylate cyclase [21]. This leads to a phosphorylation cascade that activates the steroidogenic acute regulatory protein and other multiple sequential enzymatic steps involving different cytochrome P450 enzymes and 11BHSD [22]. The zona fasciculata produces GCs, mainly cortisol, by expressing the 17α hydroxylase enzyme (CYP17A1), which mediates the conversion of 17-deoxy-21-carbon steroids into 17-hydroxy-21-carbon steroids that subsequently undergo 21-hydroxylation by the 21α hydroxylase enzyme to form 11-deoxycortisol and finally 11b-hydroxylation by the 11β hydroxylase enzyme to form cortisol [22,23].

Once GC are synthesized and secreted, they are then metabolized in peripheral tissues to their active form. First, GC availability and action depend on tissue-specific intracellular metabolism by 11BHSD. Two isoforms of this enzyme have been described. The 11β hydroxysteroid dehydrogenase type 1 isoform (11BHSD1) is expressed in key metabolic tissues such as liver, adipose tissue and skeletal muscle, and its function is to convert inactive cortisone to active cortisol. The 11BHSD type 2 isoform (11HSD2) is expressed by kidney, colon and salivary glands and catalyzes the inactivation of cortisol to cortisone, thus not only protecting the mineralocorticoid receptor from the occupancy of cortisol but also providing the substrate for 11BHSD1 in peripheral tissues [24] (Figure 1). Secondly, GC interact with their receptors. Indeed, circulating cortisol crosses the phospholipid bilayer of the cell membrane and, once inside the cell, it binds the GR, being then transferred into the cell nucleus and ultimately acting on gene transcription [22].

The GR is a member of the family of nuclear receptors, a group of ligand-dependent transcription factors [16]. Its gene, NR3C1 located on chromosome 5q31, consists of ten exons and may exist in multiple isoforms generated by alternative splicing and eight alternative initiation sites. Each GR isoform may be submitted to various posttranslational modifications, such as phosphorylation or sumoylation [25]. The GR protein is a modular protein consisting of the N-terminal transactivation domain, the central DNA binding domain and the C-terminal ligand-binding domain [16].

Following cortisol binding, a conformational change occurs that leads to the dissociation of the receptor from a large complex of proteins, of which heat shock protein 90 (HSP-90) is the most important. This activated ligand-bound receptor translocates to the nucleus, where it can initiate transcription through binding to GC-responsive elements of the target gene or affect gene transcription through direct protein–protein interaction, thus activating or repressing the target gene expression [26]. Nongenomic effects of GC have been also reported, which are mediated by other signaling pathways such as mitogen-activated protein kinase, phosphatidyl-inositol 3 phosphate, Ca^2+^/Calmodulin dependent protein kinase II or phospholipase C pathways [25].

### 3.2. Pathophysiology of Pituitary Mild Hypercortisolism

A summary of the prevalence, clinical characteristics and origin of mHC compared with Cushing’s syndrome is reported in Table 1.

By definition, pituitary-related mHC does not present with the classic features of the pituitary ACTH-dependent hypercortisolism known as Cushing’s disease [27]. The condition of pituitary mHC, because of its clinically silent nature, must not be confused with the condition of silent corticotroph adenomas, which, at variance, are defined as ACTH-expressing pituitary tumors that lack both the clinical symptoms of Cushing’s syndrome and evidence of autonomous ACTH secretion [28]. These tumors, in general, have aggressive behaviors with higher prevalence of giant adenomas and more frequent cavernous sinus invasions than other nonfunctioning pituitary adenomas [28]. In contrast, pituitary adenomas determining mHC seem to be associated with apparently nonaggressive tumors [12], but it is important to underline that, to date, there are very few data about the histological characteristics of this type of tumor. 

Whether or not differences exist between pituitary-related mHC pathogenesis and Cushing’s disease pathogenesis is still largely unknown. Ebisawa and coauthors suggested a correlation between impaired GC action and the proliferation and development of pituitary macroadenomas causing mHC, but this hypothesis has still to be demonstrated [29]. A main characteristic of the pathogenesis of pituitary ACTH-secreting adenomas is an impaired ACTH suppression by GC negative feedback, potentially caused by different molecular abnormalities. It was reported that ACTH-secreting pituitary tumors show enhanced 11HSD2 gene expression but suppressed 11HSD1 gene expression, suggesting a partial or weak GC resistance [30]. Other mechanisms possibly underlying GC resistance may be due to abnormalities in the activation and function of GR. The increased expression of the chaperone protein HSP90 that regulates GR intracellular trafficking, folding, maturation and activation has been described in these tumors. 

Testicular receptor 4, an orphan nuclear receptor, has been shown to affect the binding of GR to the proopiomelanocortin promoter, and, thus, an increased expression of this molecule is thought to play a role in causing GC resistance in ACTH-secreting adenomas. In addition, in 55% of these tumors, a loss of expression of CABLES1, a GC responsive cell cycle regulatory gene, has been reported, resulting in impaired GC sensitivity. Additionally, Brg1, which belongs to the SWitch/Sucrose NonFermentable proteins (a group of proteins that regulate the way DNA is packaged) and histone deacetylase 2 have been shown to be important for the proopiomelanocortin gene expression. Loss of either Brg1 or histone deacetylase 2 can lead to GC resistance in ACTH-secreting adenomas, since Brg1 stabilizes the interaction between GR and histone deacetylase 2 to suppress proopiomelanocortin gene transcription [30]. In summary, increased expression of the 11HSD2 gene, HSP90 or Testicular receptor 4, and loss of expression of Brg1gene or CABLES1 gene are thought to contribute to the pathogenesis of ACTH-secreting pituitary tumors by reducing the GC negative feedback system. 

A further pathogenic mechanism for ACTH-secreting adenoma tumorigenesis is represented by the increased transactivation of the proopiomelanocortin gene. Indeed, the 20–60% of ACTH-secretion tumors show mutations of the ubiquitin-specific-protease 8 gene which increase the deubiquitylation activity of this enzyme, with the ultimate effect of increasing the transactivation of the proopiomelanocortin gene. Mutations of v-Raf murine sarcoma viral oncogene homolog B and of the tumor protein P53 have been suggested to potentially play a role in ACTH-secreting tumor pathogenesis. The former mutation causes increased cell proliferation due to the activation of the mitogen-activated protein kinases, which was suggested to induce ACTH synthesis, leading to autonomous ACTH secretion in 7% of patients with Cushing’s disease. Loss-of-function mutations in the tumor protein 53, a well-known tumor suppressor gene, have been described in about 12.5% of ACTH-secreting adenoma cases. 

The above mentioned mechanisms and other factors, including arginine vasopressin and interleukin-1β, interleukin-6 and leukemia inhibitory factor, which have been recently described as having a potential role in the pathogenesis of ACTH-secreting adenomas, have been discussed in a comprehensive review by Fukuoka and coauthors [30].

In the past years, stressful conditions in early life have been found to be associated with the occurrence of Cushing’s syndrome in adult life [31,32], suggesting that conditions of chronic stress might alter the hypothalamus–pituitary–adrenal axis activity, thus leading to autonomous pituitary ACTH secretion. This hypothesis was recently investigated in a mouse model. After being stressed in early life, the mice developed an autonomous pituitary ACTH secretion with GC excess persisting even after the resolution of the stressful period. This was confirmed by high ACTH and corticosterone levels and, clinically, by a significant increase in body weight and fasting plasma glucose levels [33].

In general, stressful conditions in early life with permanent changes at pituitary and/or adrenal levels in adult life would suggest a relevant role of epigenetic mechanisms. The link between ACTH-dependent hypercortisolism and stressful conditions has been studied, for example, in type 2 diabetic patients. Type 2 diabetic patients have been shown to have increased hypothalamus–pituitary–adrenal axis activity and hypercortisolism possibly due to impaired negative feedback of the hypothalamus–pituitary–adrenal axis (GC resistance). Indeed, chronic hyperglycemia may increase the production of reactive oxygen species and oxidative stress, which may impair GR function, thus inducing a GC resistant state in the hypothalamus–pituitary–adrenal axis. In turn, chronic hypercortisolism, by worsening glucose metabolism, may increase the production of reactive oxygen species, thus perpetuating a vicious cycle [34]. On the other hand, chronic hypothalamus–pituitary–adrenal axis activation exerts a stimulating effect on cortisol-producing adrenocortical cells. Whether or not the increased prevalence of cortisol-secreting adrenal adenomas in type 2 diabetes could be due to a long-lasting ACTH hypersecretion remains an interesting hypothesis [35].

### 3.3. Pathophysiology of Adrenal Mild Hypercortisolism

Most data on adrenal mHC derive from studies on adrenal incidentalomas, the prevalence of which in subjects after 50 years of age is not negligible, estimable to be between 0.2 and 2%. Since the condition of mHC of adrenal origin is present in 5–30% of patients with AI, in individuals after 50 years of age the mHC prevalence may be estimated to be up to 0.6% [1,2,3,4,5]. 

Adrenal incidentalomas may be bilateral or unilateral. Bilateral adrenocortical tumors are characterized by the presence of several nodules in both adrenals. Their bilateral nature suggests a possible underlying genetic predisposition [23,36]. Compared with unilateral adrenal tumors, bilateral tumors are thought to be less frequent but to be more likely related to mHC, which was present in 22–42% of patients with bilateral adrenal adenomas [1,2,3,4].

Bilateral adrenocortical hyperplasia is characterized by multiple adrenocortical nodules, and it is subdivided into micronodular adrenocortical hyperplasia (nodules less than 1 cm in diameter) and bilateral macronodular adrenocortical hyperplasia (nodules greater than 1 cm in diameter) [37]. The main mechanisms involved in the pathogenesis of micronodular and macronodular adrenal hyperplasia are summarized in Table 2. Micronodular adrenal hyperplasia is rare, and primary pigmented nodular adrenal disease is its most frequent form, which is frequently related to germline protein kinase A regulatory subunit 1-alpha (PRKAR1A) mutations and associated with clinically overt cortisol excess [38,39].

On the contrary, bilateral macronodular adrenocortical hyperplasia can cause variable degrees of cortisol excess [36]. Recent advances on the pathogenesis of bilateral macronodular adrenocortical hyperplasia were made after the identification of germline mutations of the Armadillo Repeat Containing 5 (ARMC5) in about 30% of apparently sporadic cases [40,41]. The Armadillo Repeat is a tandemly repeated sequence motif (42 amino acids long) and it is involved in mediating protein–protein interactions, in the maintenance of tissue integrity and in tumorigenesis. In patients with ACTH-independent Cushing’s syndrome and in those with adrenal mHC, the prevalence of ARMC5 mutations is about 40% and 11%, respectively [40]. 

There is evidence that ARMC5, by acting as a tumor suppressor, may increase apoptosis and that its function is related to the Wnt pathway [40]. Indeed, β-catenin and adenomatous polyposis coli—two proteins containing armadillo repeats—are known to be crucial for the canonical Wnt pathway, which has been associated with adrenal tumorigenesis in functional studies. In keeping with this, the ARMC5 gene was shown to induce apoptosis in adrenal carcinoma cell lines [41]. Moreover, the ARMC5 mutations are thought to decrease the expression of steroidogenesis enzymes. The chronic steroidogenesis inhibition could induce the enlarging of the adrenal masses with consequent cortisol hypersecretion. This could explain the usually indolent and slowly progressive course leading to late development of mHC, which occurs when the adrenals are massively enlarged with multiple nodules [40]

In some patients with bilateral macronodular adrenal hyperplasia, steroidogenesis could be driven by activation of overexpressed aberrant G protein-coupled receptors, which lead to an abnormal stimulation of cortisol biosynthesis after being stimulated by their specific ligands [10]. Numerous aberrant receptors have been implicated in the pathogenesis of cortisol overproduction, including those for gastric inhibitory polypeptide, vasopressin, angiotensin type 1, luteinizing hormone/human chorionic gonadotropin, serotonin and glucagon [10]. These aberrant receptors have been identified in up to 50% of patients with unilateral adenomas associated with hypercortisolism, with higher prevalence in those patients with mild hypercortisolism [10,42].

Recent advances have been made in understanding the genetic background of unilateral adrenal adenomas, as summarized in Table 3. As for bilateral adrenal tumors, somatic alteration of actors of the cAMP/PKA pathway were found in unilateral cortisol-secreting adenomas. Mutations of the catalytic alpha subunit of the PKA enzyme (PRKACA) are often present in patients with unilateral adenomas and are related to clinically overt Cushing’s syndrome [43,44,45,46,47,48,49,50], while they have been rarely (less than 5% of cases) detected in patients with mHC [10]. At variance, whole exome sequencing investigations revealed a high prevalence of mutations of CTNNB1, the gene encoding β-catenin, in patients with mHC. The idea that the pathogeneses of adrenal Cushing’s syndrome and adrenal mHC are substantially different was proposed in 2013 by Roussel and coauthors, who performed a transcriptome analysis on a small series of cortisol-producing adenomas, which showed an association between cortisol secretion and expression of subsets of genes implicated in steroid secretion, potentially different between Cushing’s syndrome and mHC [51]. These authors also showed that the phosphodiesterase 8 (PDE8B), which inactivates the PKA pathway, had the strongest positive correlation with cortisol secretion. In adenomas with high PDE8B levels, the PKA/cAMP ratio was increased, representing possible negative feedback to limit the pathway hyperactivation [51]. In a subsequent study, Ronchi and coauthors, by analyzing the genetic landscape in 99 adrenal tumors, found a total of 706 somatic protein-altering mutations in 88 samples. Interestingly, the genetic alterations in different genes involved in the Wnt/β-catenin pathway were associated with larger tumors and endocrine inactivity [49]. Notably, mutations were highly frequent in many genes of the Ca2^++^-signaling pathway, particularly those encoding ryanodine receptors, which are intracellular Ca2^++^ release channels exerting a potential role on adrenal function [52]. Indeed, the adrenal fasciculate zone expresses high levels of Ca2^++^ channels possibly involved in the regulation of cortisol secretion [53], and Ca2^++^ channels have been shown to be possibly involved in molecular mechanisms of apoptosis regulation and cancer transformation [54], suggesting a possible role of this pathway in the adrenocortical cells proliferation.

Interestingly, in the paper of Di Dalmazi and coauthors, unsupervised clustering showed a separation of patients with Cushing’s syndrome from those with mHC and nonfunctioning adrenal tumors. The idea of a different pathophysiological background between adrenal Cushing’s syndrome and adrenal mHC was recently reinforced by a study investigating the association between transcriptome and mutational status of benign adrenocortical tumors [55]. In this study on frozen tissue of 59 adrenal tumors with known genetic background, the authors showed that that transcriptome analysis (gene expression, long noncoding RNA expression and gene fusions) identified two different clusters for adrenal adenomas, the first one mainly consisting of adenomas with mHC and not functioning adrenal adenomas with CTNNB1 or without identified driver mutations, the second one including adenoma with Cushing’s syndrome with or without identified driver mutations. Therefore, it is possible to hypothesize that molecular alterations in the cAMP/PKA or Wnt/beta catenin pathways might be involved in the pathogenesis of adrenocortical tumors, with the mechanisms underlying mHC being potentially different from those determining Cushing’s syndrome [55]. 

In summary, the presence of a potential genetic cause has been found in about 40% of patients with mHC [23,40,43,44,45,46,47,48,49] (5% PRKARCA mutations, 25% CTNNB1 mutations and 11% ARMC5 mutations), while, in about 60% of patients, possible genetic alterations have not been shown yet and epigenetic mechanisms may play a role. Among the epigenetic changes, DNA methylation is one of the most studied regulatory mechanisms. Nowadays, a debated issue is whether or not DNA methylation may be a putative regulatory mechanism of 11BHSD1 (the enzyme catalyzing the transformation of deoxy-cortisol in cortisol) in cortisol-producing adenomas. Indeed, while Kometani and coauthors showed a lower DNA methylation level and a higher mRNA expression of 11BHSD1 in cortisol-producing adenomas, as compared with nonfunctioning adenomas [56]. Di Dalmazi and coauthors did not find differences in methylation patterns in 11BHSD1 between these two groups [57]. Finally, very recently, alterations of DNA methylation were suggested to be possibly associated with the growth and worsening of cortisol secretion in pregnancy [58].

An important aspect of adrenal mHC pathophysiology is related to the pattern of adrenal steroids secretion. Several studies have analyzed AI hormonal secretion, evaluating the amount of secreted GC and mineralocorticoid precursors, with AI with mHC often showing the cosecretion of different steroids [59]. Interestingly, AI with biochemically demonstrated mHC have been shown to have steroid secretions profoundly different from that of AI with Cushing’s syndrome [60]. In the old study by Rossi and coauthors, the most common endocrine abnormalities in patients with AI were high 17-hydroxyprogesterone and low dehydroepiandrosterone-sulphate levels [45]. In a more recent study, conversely, Di Dalmazi and coauthors, by studying steroid profile by liquid chromatography–tandem mass spectrometry (LC–MS/MS), found significantly decreased levels of DHEA and androstenedione in patients with mHC [61]. The decreased androgens levels in patients with mHC can be explained by different hypotheses. First, it may be attributable to a dysregulation of CYP17A1 and, thus, of 17-hydroxylase and 17,20-lyase activities, which could be attributable to an intrinsic intra-adrenal adaptor mechanism to limit the androgens overproduction [59]. On the other hand, the suppression of ACTH, driven by the increased cortisol secretion by the adenoma, can lead to decreased stimulation of the adjacent and contralateral adrenal cortex, resulting in low androgen production [3]. This latter hypothesis is also supported by the fact that in AI patients, the ACTH levels are directly associated with DHEA and androstenedione levels [61]. Recently, the androgen steroid profile including 11-deoxycortisol, 11-deoxy-corticosterone and corticosterone was suggested to be as accurate as the dexamethasone suppression test in identifying mHC in patients with AI [60].

GC exposure is thought to be influenced by the different GR polymorphisms [16]. Database for Single Nucleotide Polymorphisms (SNPs) currently lists 3016 active single nucleotide polymorphisms in the human GR gene, most of them in introns or untranslated regions and with frequencies of the minor allele well below 1% [16]. In several but not all studies, polymorphisms in the GR gene seem to correlate significantly with variation of sensitivity to endogenous GC in normal individuals [25]. Some data suggested that some GR gene variants may have a role in the pathogenesis of adrenal tumors. In a previous study, the BclI variant was associated with the AI presence [62]. In subsequent studies, the BclI single nucleotide polymorphism was found to be associated only with small AI [63], while the prevalence of the N363S GR variant was shown to be markedly higher in patients with bilateral AI, suggesting a possible pathogenic role [64]. As far as Cushing’s syndrome is concerned, data are very discordant. Indeed, while a study showed a higher BclI variant prevalence in Cushing’s syndrome [65], other data did not confirm this association, showing that the GR gene nucleotide polymorphisms BclI, N363S, ER22/23EK and A3669G did not influence the development of both pituitary and adrenal Cushing’s syndrome [66,67].

## 4. Peripheral Glucocorticoid Sensitivity and Activation: From Eucortisolism to Mild Hypercortisolism

After their synthesis and secretion, GC are metabolized in peripheral tissues to their active forms and interact with their receptors. The overall GC effects on target tissue are different among the different individuals and among the different tissues in the same individual, and this may be due to several mechanisms [4]. The term “GC tissue exposure” summarizes the effects of the type and amount of secreted GC, of the degree of 11BHSD1 activity and of the GR genetic variants. The role of 11BHSD1 in the pathogenesis of hypercortisolism was first studied by Tomlinson and coauthors, who reported a rare case of a patient with pituitary-dependent hypercortisolism, who appeared to be protected from the classical Cushing’s phenotype. Specifically, this patient had normal fat distribution, absence of myopathy and normal blood pressure. Subsequent investigation revealed a functional defect in 11BHSD1 activity, as evidenced by serum and urinary biomarkers [68]. In more recent years, Morgan and coauthors studied the metabolic effects of 11BHSD1 activity in different tissues on transgenic animal models with targeted deletions of 11BHSD1. The study concluded that adipose-specific 11BHSD1 knockout mice were protected from hepatic triglyceride accumulation, increased serum fatty acids and increased expression of adipose lipolytic enzymes, while liver-specific knockout mice were not protected from any of these GC-dependent effects. The liver-specific knockout mice were not protected from glucose intolerance, hyperinsulinemia, decreased lean mass or increased adiposity. This study demonstrated that in mice, GC reactivation by 11BHSD1 in peripheral tissues is a major determinant of the phenotype of GC excess, thus representing a potential therapeutic target [15,24]. 

However, few studies have linked an overexpression or overstimulation of 11BHSD1 with the typical metabolic alterations of metabolic syndrome in the absence of biochemical evidence of cortisol excess. Masuzaki and Flier developed a strain of transgenic mice with an adipocyte-specific overexpression of 11BHSD1 which showed a phenotype similar to the metabolic syndrome [69]. Furthermore, in vitro data strongly suggested that the accumulation of a series of humoral factors, which are known to be implied in the metabolic syndrome, such as tumor necrosis factor-α and interleukin-1B, have a stimulatory effect on the 11BHSD1 transcriptional activity, which increases the local concentration of active cortisol and causes an “intracellular Cushing state” in the liver, leading to the occurrence of metabolic disorders such as glucose intolerance and/or abnormal lipid metabolism [70]. In keeping with these animal studies, we found that in postmenopausal eucortisolemic subjects, the 11HSD1 activity (as evaluated by the urinary free cortisol/urinary free cortisone ratio) was directly associated with the risk of being affected by two of the following: type 2 diabetes, hypertension and bone fragility [17,18].

Besides 11BHSD enzymes, GC exposure is thought to be influenced also by the above mentioned different GR gene polymorphisms [16]. In particular, GR gene variations have been found to be associated with different phenotypes in body composition, the cardiovascular system, the immune system and psychiatric disorders [26]. The ER22/23EK (rs6189 and rs6190) polymorphism, which is found in 2% of subjects in the general population, is associated with decreased GC sensitivity. The presence of the ER22/23EK polymorphism results in a healthier metabolic profile, including lower total cholesterol levels, lower fasting insulin levels, increased insulin sensitivity and a lower risk of type 2 diabetes mellitus [16,26]. This polymorphism is also associated with a lower risk of developing dementia in elderly patients [25]. Secondly, the GR-9Beta (rs6198 or A3669G) polymorphism, with an 8% prevalence in the general population, is also associated with decreased sensitivity. The presence of this single nucleotide polymorphism was shown to be associated with a decreased risk of obesity in women and with decreased total cholesterol levels and increased HDL cholesterol levels [25]. On the other hand, the BclI polymorphism (rs41423247), which is reported in 25% of the general population, is associated with increased GC sensitivity. Various studies linked this polymorphism to higher body mass index, higher waist circumference, higher fasting glucose and insulin levels in men [25]. The BclI polymorphism was also associated with the presence of major depression in a large Caucasian population [25]. Finally, the sensitizing N363S (rs6195) polymorphism, found in 2.3% of the general population, was linked to significantly higher body mass index and higher risk of hypertension [25]. In keeping with the possible effect of GR single nucleotide polymorphisms on GC target tissue in eucortisolemic subjects, in postmenopausal women without cortisol excess the N363S GR single nucleotide polymorphism was directly associated with the risk of being affected by two of the following: type 2 diabetes, hypertension and bone fragility [18].

On the basis of these data, it is conceivable that, not only in the general population but also in patients with mHC, the presence of different 11BHSD1 activities and of different GR single nucleotide polymorphisms could determine a different GC exposure, that, for the same GC secretion, may lead to profoundly different frequencies and severities of GC-related chronic consequences. While in Cushing’s syndrome the huge amount of secreted GC overcomes the inactivating activity exerted by the 11BHSD2 enzyme and renders the different 11BHSD1 and GR single nucleotide polymorphisms uninfluential, in mHC these latter factors of GC exposure could become crucial. This is suggested by the finding that in AI patients, the 11BHSD2 activity was found to be directly associated with cortisol secretion in patients with mHC bot not in those with Cushing’s syndrome [71].

Nowadays, however, data regarding the possible role of the different 11BHSD1 activity and of the different GR single nucleotide polymorphisms in mHC are controversial. Indeed, in a previous study we showed that in AI patients, the presence of the sensitizing haplotype of BclI and N363S was associated with the concomitant presence of arterial hypertension and morphometric vertebral fractures [71]. Partially at variance, more recently, in a large sample of AI patients (*n* = 411) Reimondo and coauthors found that the N363S GR single nucleotide polymorphism was associated with hypertension but only in apparently nonfunctioning AI [67]. Data regarding the role of GC exposure in the different GC-related complications in mHC will be summarized in the following paragraphs (Figure 2).

## 5. Pathophysiology of the Systemic Consequences of Mild Hypercortisolism

### 5.1. Bone Fragility in Mild Hypercortisolism

As constantly reported in patients with Cushing’s syndrome and even in patients with mHC, bone density and bone quality are altered. The prevalence and incidence rates of vertebral fracture in people with hypercortisolism are extremely high and could represent a large burden of the disease that is currently not being addressed [72]. A meta-analysis by Chiodini and coauthors reported that the prevalence of radiographically identified vertebral fractures was 63.6% in patients with mHC compared with 16% in controls [73]. 

The direct effects of GC excess on bone cells are characterized by reduced bone apposition due to an impairment of osteoblasts and osteocyte function (Figure 3) [74].

Indeed, GC reduce the osteoblastogenesis from the mesenchymal stem cells by directing their differentiation to adipocytes rather than osteoblasts [75,76,77]. The mechanisms involved in this redirection of mesenchymal cells include the induction of the nuclear factors of the CCAAT/enhancer binding protein family and of the peroxisome proliferator-activated receptor γ2, as well as the inhibition of Wnt/β-catenin signaling (Figure 4). 

Cortisol excess also induces differentiation and maturation of osteoclasts (Figure 3). The examination of the role of osteoclasts is made more difficult by the fact that GC have direct effects on osteoclasts and their precursors and also powerful indirect influences on osteoclastogenesis and osteoclast function via effects on osteoblasts and osteocytes [72]. Indeed, GC also affect the function of osteocytes, thus impairing the bone biomechanical properties [72].

The indirect effects of GC on bone are mediated by their inhibition of the gonadotropins release, which, in turn, affects estrogen and testosterone production [73]. It is also well-known that GC plays a role in the inhibition of the production of growth hormone and insulin-like growth factor, which are important stimulators of osteoblasts [74]. Calcium homeostasis is also adversely affected by GC. The intestinal capacity to absorb calcium is lowered and calcium reabsorption by the renal tubule is inhibited by GC [73].

Finally, GC exposure has been investigated as a possible contributor to different bone phenotype in patients with endogenous cortisol excess. Particularly in patients with mHC, in whom the amount of secreted GC is not dramatically increased, a different pattern of GC peripheral activation could play a significant role [73]. In bone, 11BHSD1 activity was shown to be important in bone formation, and of utmost importance, osteoblastic 11BHSD1 activity was demonstrated to increase with age and GC exposure. Autocrine GC activation within bone cells is thought to influence the age-related decrease in bone formation and the increased fracture risk in patients with glucocorticoid-induced osteoporosis [73,75]. Consistently, in a mouse model, inhibiting the 11BHSD1 activity was shown to enhance the osteoblastogenesis and inhibit the osteoclastogenesis [76]. More recently in a study on mouse bone-like cells (MLO-Y4) and mouse osteoblast-like cells (MC3T31), after overexpressing 11BHSD2, the authors induced cell apoptosis by dexamethasone. They found that 11BHSD2 overexpression decreased the apoptosis of MC3T3/MLO-Y4 cells induced by GC [77]. In line with this finding, in a mouse model, Fenton and coauthors demonstrated that 11BHSD1 mediates GC suppression of anabolic bone formation and reduced bone volume secondary to a decrease in osteoblast number [78].

Some evidence in humans seems to confirm the results of cellular and animal studies. Indeed, in a cohort of subjects of the Hertfordshire Cohort Study, circulating cortisone levels were found to be associated with biochemical markers of bone formation and lumbar spine bone mineral density [79]. Furthermore, in a small study, Hwang and coauthors showed that HSD11B1 polymorphisms could predict bone mineral density changes and fracture risk in postmenopausal women without a clinically apparent hypercortisolemia [80]. These data have been subsequently confirmed by a larger (*n* = 452) study in patients evaluated for osteoporosis which showed that the genetic polymorphisms in 11BHSD1 correlated with the postdexamethasone cortisol levels and bone mineral density [81]. The role of 11BHSD enzymes in the context of bone metabolism and bone cell function was nicely and extensively reviewed by Martin and coauthors [82].

GC sensitivity has been suggested to potentially play a role in influencing bone damage in patients with Cushing’s syndrome. In a study on 60 patients with Cushing’s syndrome, subjects carrying the BclI polymorphism in a homozygous form had reduced femoral bone mineral density at femoral subregions and higher bone resorption compared to patients with the wild-type variant [66]. As far as patients with mHc are concerned, in a sample of 72 patients with AI we found that the presence of morphometric vertebral fracture increased five-fold in the presence of either mHC or the GC-sensitizing haplotype (i.e., homozygous BclI and heterozygous N363S polymorphism) as compared with patients without mHc or the GC-sensitizing haplotype (i.e., wild-type BclI, wild-type N363S and heterozygous BclI polymorphism) [71]. Larger studies are needed to better explore the role of GR polymorphic variants in influencing the bone effects of mHC.

### 5.2. Hypertension in Mild Hypercortisolism

Arterial hypertension is present in over 60% of mHC patients [83]. At the time of diagnosis, about 58–64% of patients with mHC display hypertension [54]. It has been suggested that the physiological nocturnal blood pressure decrease could be a specific feature of cortisol-induced hypertension [84]. 

The mechanisms involved in the pathogenesis of cortisol-related hypertension are still not fully clarified. Firstly, in hypercortisolism an increased activity of the renin-angiotensin system is present. In fact, hypercortisolism leads to increased hepatic production of angiotensinogen, and in patients with hypercortisolism an upregulation of angiotensin II type 1A receptors has been described [85]. A second mechanism could be an imbalance in the vasoregulatory system due to inhibition of vasodilators and increased vascular reactivity to vasopressors, as suggested by the presence of increased endothelin 1 in patients with hypercortisolism [86]. Finally, the 11BHSDs activity and the GR could have a role. Indeed, 11BHSD1 knockout mice are protected from the effects of GC excess, including hypertension, hepatic steatosis, myopathy and dermal atrophy. Importantly, only the liver-specific 11BHSD1 knockout mice developed a full Cushingoid phenotype, while the adipose-specific 11BHSD1 knockout mice were protected [15]. It is known that in Cushing’s syndrome the 11BHSD2 capacity to inactivate cortisol is overwhelmed by the huge amount circulating cortisol. Once delivered to key metabolic target tissues, cortisone is reactivated to cortisol by 11BHSD1 to allow GR activation. Interestingly, a possible interaction between 11BHSD2 activity and GR activity was suggested by the presence of GR-dependent stimulation of 11BHSD2 catalytic activity in a study on 6 patients with loss-of-function GR gene mutations. In these subjects hypercortisolism was associated with low potassium levels and low plasma renin and aldosterone levels, regardless of hypertension [87]. Overall, these data suggest that, at variance with overt GC excess, in which the huge amounts of secreted cortisol generally overcome the possible influence of GR single nucleotide polymorphisms and/or 11BHSDs activity, in patients with mHC displaying a less severe cortisol excess, the different GR single nucleotide polymorphism and 11BHSD activity may play a role. Further studies are warranted for clarifying this hypothesis.

### 5.3. Subclinical Atherosclerosis, Cardiovascular Remodeling and Coagulation in Mild Hypercortisolism

Patients with AI and mHC have a higher risk of cardiovascular events (including coronary artery disease, myocardial infarction, stroke, transient ischemic attack and heart failure) and mortality compared to patients with nonsecreting AI and controls [83,88]. Importantly, the increased cardiovascular risk and mortality in mHC is independent of the presence of diabetes and hypertension [89,90,91,92].

In patients with Cushing’s syndrome, the presence of left ventricular hypertrophy, the decrease in systolic strain and the impaired relaxation pattern with a decrease in diastolic filling were reported in several studies. The possible mechanisms underlying these alterations include an enhanced response to angiotensin II and a GC-induced activation of the mineralocorticoid receptor, leading to myocardial fibrosis, ventricular remodeling, impairment of relaxation and, eventually, heart failure [84]. 

Finally, the possible role of accelerated subclinical atherosclerosis in the determination of cortisol excess-related CV risk should not be overlooked. Several studies reported an increased intima–media thickness, higher prevalence of carotid plaques and lower flow-mediated dilatation in patients with overt Cushing’s syndrome compared to controls [83]. Interestingly, in a study including 60 normotensive euglycemic patients with AI and 32 healthy controls with normal adrenal imaging, the intima–media thickness correlated with cortisol levels after DST, whose cutoff set at 1.4 μg/dL had higher accuracy (sensitivity 79.2%, specificity 88%) in predicting the cardiovascular risk profile [93]. This study confirms previous evidence that several cardiometabolic risk factors occur with higher prevalence of mHC compared to age-matched healthy subjects [94,95,96,97].

The alterations of coagulation could possibly explain the increased cardiovascular risk in mHC. However, while a significant body of research on the coagulation system confirms the increased thromboembolic risk in overt Cushing’s syndrome patients [98], much less evidence is available on thromboembolic complications in mHC subjects [99]. Hypercortisolism significantly increases the risk of dangerous thromboembolic events through its multilevel prothrombotic effect on hemostasis [99]. Epidemiological studies show several-fold greater incidences of thromboembolic events in hypercortisolemic patients compared to those without hormonal disorders, even though the issue of discriminating between the effects of GC and the effects of obesity and organ damage is still open [98]. The prothrombotic state is caused by both activation of the coagulation system and inhibition of fibrinolysis. In hypercortisolemic subjects, the typical hemostatic alterations are a rise in factor VIII, fibrinogen and von Willebrand factor levels as well as a shortening of the activated partial thromboplastin time. Additional alterations are an increase in the number of platelets, thromboxane B2 and thrombin–antithrombin complexes [98].

In mHC, attention has been focused on the activity and concentration of von Willebrand factor, as GC modulate von Willebrand factor gene transcription [99]. Indeed, von Willebrand factor was reported to be higher in adrenal mHC compared to controls, but no statistically significant difference was observed in the levels of other coagulation factors [100]. Patients with mHC also have higher levels of protein C, protein S and thrombomodulin. These disturbances in the endogenous anticoagulation system could be a compensatory response to the prothrombotic state [101].

### 5.4. Lipid and Glucose Metabolism in Mild Hypercortisolism

Dyslipidemia is described in 55–71% patients with mHC [3]. The effect of mHC on lipid profile may be indirect, as type 2 diabetes, insulin resistance and weight gain contribute to dyslipidemia. Some data suggest that adrenalectomy might result in an improvement of dyslipidemia in patients with adrenal mHC, although the effect of recovery from mHC is not as robust as for hypertension and type 2 diabetes [3]. Furthermore, in a previous study on 334 AI patients, we found that in the absence of alterations of glucose metabolism the mHC presence had no effect on lipid pattern, while the impaired glucose metabolism influenced the lipid metabolism regardless of the mHC presence [102]. 

The pathogenesis of dyslipidemia in hypercortisolism includes direct and indirect cortisol effect on lipolysis, free fatty acid production and turnover, very-low-density lipoprotein synthesis and fat accumulation in the liver [83,103] and usually presents with low HDL and elevated LDL and triglyceride concentrations [84]. In addition, modulation of GR activation plays a possible role in the pathogenesis of dyslipidemia [103]. It was demonstrated that both hepatic overexpression of 11BHSD1 and GR genetic variants in the liver contributed to the development of hepatic steatosis [89].

Chronic exposure to exogenous GC as well as overt hypercortisolism is associated with an increased risk of developing type 2 diabetes [83]. The type 2 diabetes prevalence in mHC is largely variable among the different studies due to the different procedures (i.e., fasting glucose vs. oral glucose tolerance test) used for diagnosing type 2 diabetes as well as the difference in the cutoffs applied for mHC diagnosis. Overall, it is estimated that one-third of patients with mHC suffer from type 2 diabetes [102].

GC are diabetogenic agents because they cause a beta-cell dysfunction and reduce insulin sensitivity [104]. The beta-cell dysfunction is caused by the binding between GC and GR in pancreatic beta-cells which induces an impairment of uptake and metabolism of glucose. The insulin resistance is caused mainly by impaired glucose transporter migration to the cell surface. In fact, GC excess induces a postreceptor defect of insulin receptor substrate-1, phosphatidylinositol-3 kinase and protein kinase B in the liver, skeletal muscle and adipose tissue. Insulin resistance is also indirectly favored in hypercortisolism by the increased proteolysis and lipolysis resulting in elevation of amino acids and fatty acids, which in turn may cause impairment in different steps of insulin signaling [104]. Thus, GC excess results in impaired glucose metabolism primarily through a decrease in insulin action and reduction in glucose disposal. The compensatory increase in insulin secretion is not sufficient to compensate for the significant alteration in insulin receptor signaling in the liver and peripheral tissues [105]. 

These mechanisms are likely to explain the occurrence of type 2 diabetes even in mHC. In this regard, the level of GC excess needed to induce type 2 diabetes occurrence is a topic of interest. In a study including 60 normotensive euglycemic patients with AI and 32 healthy controls with normal adrenal imaging, the cutoff of cortisol after DST set at 1.1 μg/dL had higher accuracy (sensitivity 87.5%, specificity 88%) in predicting insulin resistance [93]. In keeping with the idea that cortisol after a DST cutoff set at 1.8 μg/dL (as currently recommended) may be too high to predict the presence of type 2 diabetes in AI patients, in a 3-year follow-up study on 166 patients with AI and 740 subjects without AI, participants with nonsecreting AI (i.e., with cortisol after DST < 1.8 μg/dL) had a 1.87 times higher risk for incident type 2 diabetes than those without AI. These data suggest that the current classification of AI as “nonfunctional” should be reassessed.

Other aspects potentially contributing to the pathogenesis of type 2 diabetes in mHC are GC’s influence on the adipokines secretion, such as adiponectin and leptin, which may have a relevant impact on insulin sensitivity [104] and GR genetic variants. Indeed, adiponectin levels were found to be greater in patients with AI and mHC than in controls [106] and the GR genetic variant A3669G was found to be associated with a reduced the risk of developing type 2 diabetes in patients with Cushing’s syndrome [65]. The authors speculated that the isoforms of GR may play a role in type 2 diabetes pathogenesis in Cushing’s syndrome. Indeed, the effects of GCs are mediated by the isoform GR-a, while the isoform GR-b is thought to have a dominant-negative inhibition on GR-a and contribute to GC resistance [107]. The A3669G GR polymorphism presence could result in increased GR-b expression, leading to a greater inhibition of GR-a transcriptional activity and consequently to a reduced GC sensitivity [107].

### 5.5. Skeletal Muscle in Mild Hypercortisolism

In patients with Cushing’s syndrome, myopathy is highly frequent (prevalence 42–83%), particularly in those with ectopic Cushing’s syndrome and in males. Hypercortisolism-related myopathy most commonly, but not exclusively, affects the proximal part of lower limbs. GC excess leads to type 2 muscle fibers atrophy through both anti-anabolic and catabolic actions (Figure 5). 

GC excess leads to type 2 muscle fibers atrophy through both anti-anabolic and catabolic actions. The former mechanisms include the decrease in amino acid uptake by muscle; the inhibition of insulin-like growth factor 1 secretion, which is crucial for the rapamycin pathway and the inhibition of myogenin, which, in turn, is essential for myogenesis and the stimulation of myostatin. In muscle, the mammalian target of rapamycin, known as mTOR, is the central controller of protein synthesis, and its inhibition is a crucial mechanism for GC excess-related sarcopenia. Myostatin is a myokine acting in an autocrine manner to suppress skeletal muscle growth, and it is thought to play a role in reducing insulin sensitivity. Thus, the increased myostatin levels could possibly contribute to both myopathy and impaired glucose metabolism in Cushing’s syndrome. GC excess enhances proteolysis, due to alterations of the ubiquitin–proteosome system and to impairment of sarcolemma excitability. In particular, GC-induced protein degradation seems to be related to GC effects on the muscle-ring finger protein-1 (MuRF1) and atrogin-1, both target cellular proteins to the proteosome for hydrolysis. Indeed, by linking to their GC receptor, GC lead to upregulation of forkhead box O3a (FOXO3a), MuRF1 and atrogin-1 mRNA, ultimately increasing muscle atrophy. Interestingly, the GC excess-induced decrease in insulin-like growth factor 1 levels was associated with a decrease in mTOR activation and increased mRNA expression of FOXO3a, MuRF1 and myostatin. In line with these data in a chronic GC excess model, a significant muscle fibers atrophy associated with enhanced gene expression of Murf1 and myostatin was found. Importantly, the 11BHSD1 inhibition was found to counteract the GC-induced decrease in protein synthesis and increase in protein degradation, suggesting a potential role for 11BHSD1 inhibitors to ameliorate muscle-wasting effects associated with GC excess. In line with these data, recently in a small study, the 11BHSD1 inhibitor S-707106 demonstrated an effective insulin sensitizer, anti-sarcopenic and anti-obesity effect in both Cushing’s syndrome and mHC patients [108,109,110,111,112,113].

The former mechanisms include the decrease in amino-acid uptake by muscle; the inhibition of insulin-like growth factor 1 secretion, which is crucial for the rapamycin pathway and the inhibition of myogenin, which is essential for myogenesis and the stimulation of myostatin [98,108,109]. The GC excess-mediated catabolic action on muscle is mainly related to enhanced proteolysis, due to alterations of the ubiquitin–proteosome system and to impairment of sarcolemma excitability [110], via effects on the muscle-ring finger protein-1 (MuRF1) and atrogin-1, both target cellular proteins to the proteosome for hydrolysis [108]. Interestingly, the decrease in insulin-like growth factor 1 levels due to GC excess may also have catabolic effects, since it was suggested to be associated with a decrease in mTOR activation and increased mRNA expression of MuRF1 and myostatin [108,111]. Importantly, the 11BHSD1 inhibition was found to counteract the GC-induced decrease in protein synthesis and the increase in protein degradation, suggesting a potential role for 11BHSD1 inhibitors to ameliorate muscle-wasting effects associated with GC excess [112]. In line with these data, recently in a small study, the 11BHSD1 inhibitor S-707106 demonstrated an effective insulin sensitizer, anti-sarcopenic and anti-obesity effect in both Cushing’s syndrome and mHC patients [113]. Finally, it is known that Cushing’s syndrome affects muscle strength in the acute phase, but functional impairment remains detectable also during long-term follow-up despite biochemical remission [114]. Importantly, lower IGF-I levels six months after recovery from Cushing’s syndrome were found to be associated with adverse long-term myopathy [115], and an inverse correlation between skeletal muscle mass (as evaluated by magnetic resonance imaging) in remission from Cushing’s syndrome and duration of postoperative oral GC administration was suggested [116].

Data on muscle health in mHC are scarce. Recently, Delivanis and coauthors studied body composition in 227 AI patients, including 76 and 131 patients with mHC and with nonfunctioning AI (i.e., with cortisol after 1 mg overnight DST < 1.8 μg/dL), respectively. Both groups showed a higher visceral fat and a lower skeletal muscle area. Interestingly, for every 1 µg/dL cortisol increase after DST, the visceral fat/muscle area ratio significantly increased by 2.3 times, and the mean total skeletal muscle area significantly decreased by 2.2 cm^2^. These data not only suggest that, similarly to Cushing’s syndrome patients, mHC patients are at risk of sarcopenia but also show that, at least as far as skeletal muscle is concerned, the DST cutoff set at 1.8 μg/dL is not sensitive enough to identify AI patients at risk for sarcopenia [117]. One year after adrenalectomy, skeletal muscle density measured by computed tomography was found to ameliorate, and in a multivariate linear regression model, the increase by 1 μg/dL of cortisol after DST measured before surgery was independently associated with greater reduction in visceral fat area and of visceral fat/subcutaneous fat ratio after surgery [118].

Overall, these data suggest that even mHC might determine the type 2 muscle fibers atrophy through both anti-anabolic and catabolic actions, as in the condition of Cushing’s syndrome, and also that the sarcopenic effects of mHC could be influenced by the tissue GC activation by 11HSD1, which, therefore, represents a potential therapeutic target.

### 5.6. Mood Disorders in Mild Hypercortisolism

It is well-known that neuropsychiatric diseases are severe comorbidities of Cushing’s syndrome. The most common psychiatric diseases in Cushing’s syndrome are major depression (prevalence 50–81%), anxiety (66%) and bipolar disorders (30%) [98]. Concerning cognitive functions, in Cushing’s syndrome the most frequent reported alterations are memory impairment (about 83% of cases) and reduced concentration (66% of cases) [119]. Unfortunately, low health-related quality of life and psychiatric symptoms are only partially reversible after hypercortisolism resolution [120]. Indeed, following successful treatment of Cushing’s syndrome, one-fourth of patients still experience depressed mood, and the cognitive impairments are only partially restored [121]. The central nervous system is rich in GR and chronic brain exposure to cortisol excess causes deep structural and functional changes in cerebral areas particularly rich in GR and therefore particularly vulnerable to GC excess, such as the hippocampus, which is critical for learning and memory [122].

The GC peripheral activation and sensitivity may influence even the occurrence of neuropsychologic disorders. In patients affected by Cushing’s syndrome, the 11BHSD1 genetic variants and the Bcl1 single nucleotide polymorphism of GR were found to be associated with impaired cognitive function [123]. In addition, another study reported that hypomethylation of FKBP5 gene, which encodes a cochaperone of HSP90 protein that regulates intracellular GR sensitivity, was associated with reduced hippocampal volume in patients with Cushing’s syndrome [124]. To date, only one very recent study explored the impact of mHC on mental health and cognitive function. In this study, 62 AI were evaluated. The authors found that patients with mHC had increased levels of disability related to mental illness, higher levels of perceived stress, lower levels of perceived social support and higher prevalence of middle insomnia [125].

### 5.7. Infections in Mild Hypercortisolism

The effects of GC excess on the innate immune system have been known for many years. Summarizing, macrophage maturation, monocyte production, neutrophil function and the lymphocyte natural killer action are affected by GC excess, with a consequent alteration of the cellular response to infection. Additionally, the humoral response to infection is also influenced by GC excess, which induce the downregulation of pro-inflammatory cytokines. GC excess also affect T-cell maturation and B-cell proliferation by inhibiting the antigen-presenting dendriting cells. Importantly, GC suppress the actions of the lymphocytes T-helper1, which are crucial for the production of interferon γ, interleukin 2 and tumor necrosis factor β and for the B cell production of IgG antibodies. These premises explain the 2.4–5.7 times greater susceptibility to intracellular and opportunistic infections in patients with GC excess, which represents an important cause of mortality in these patients [98].

In patients with GC excess, the susceptibility to infections was traditionally considered to be related to the degree of GC excess [126]. However, more recently Debono and coauthors retrospectively assessed overall survival in 272 AI patients and found that mortality increased twelve-fold in AI patients with mHC and that 33% of deaths were secondary to respiratory/infective causes [92]. In line with these data, a case of a fatal infection including disseminated herpes zoster triggered by pituitary mHC subclinical Cushing’s disease has been described [127]. Thus, prospectively investigating infection risks is worth performing in mHC studies.

## 6. Conclusions

The pathophysiology of mHC is receiving increased interest due to three reasons. 

Firstly, it is becoming clear that, clinically, the pathogeneses of Cushing’s syndrome and mHC are different and the latter condition could not be considered a first step or an initial stage of the former. Indeed, while for the pituitary mHC, data are still scarce, recent genetic studies have clarified some important differences between adrenal Cushing’s syndrome and adrenal mHC. These data could represent, in the future, a milestone to start from for personalizing the diagnostic procedures and therapeutic management of these patients. Given the significant prevalence of mHC in the general population, developing precision medicine in this field could lead to reducing the economic and social costs caused by the deleterious effects of the complications of mHC [128]. 

The second aspect is a direct consequence of the first. Predicting the consequences of mHC and thus addressing its treatment of choice (i.e., conservative or surgical) in the individual patient is still challenging for physicians and it is becoming a widely debated topic. Even perhaps more importantly, recent data suggest that the current tests for diagnosing mHC are not sensitive enough, possibly leaving many patients with mHC untreated [7,8,37]. These findings might push the researchers to reconsider the normal values of the different tests currently used for diagnosing hypercortisolism.

A third reason is related to potential therapeutic implications of the idea that the combination of GC secretion, GC peripheral activation and GC sensitivity is crucial for the development of mHC-related chronic consequences. In this regard, very recently, the 11BHSD1 inhibitor S-707106 was found to be an effective insulin sensitizer, anti-sarcopenic and anti-obesity drug for patients with Cushing’s syndrome and mHC [113].

Finally, we are still searching for a possible role of the degree of GC exposure in the pathogenesis of some chronic disorders. Indeed, hypertension, type 2 diabetes and mood disorders have been suggested to be potentially linked with an increased GC exposure, even in patients with apparently normal GC secretion [18,129,130,131,132,133,134,135]. Whether treating this functional hypercortisolism could be of help in curing these chronic diseases represents a frontier for medicine. To reach this goal, the mHC condition could represent an interesting model of a slight and longstanding endogenous GC excess.

## Figures and Tables

**Figure 1 ijms-23-00673-f001:**
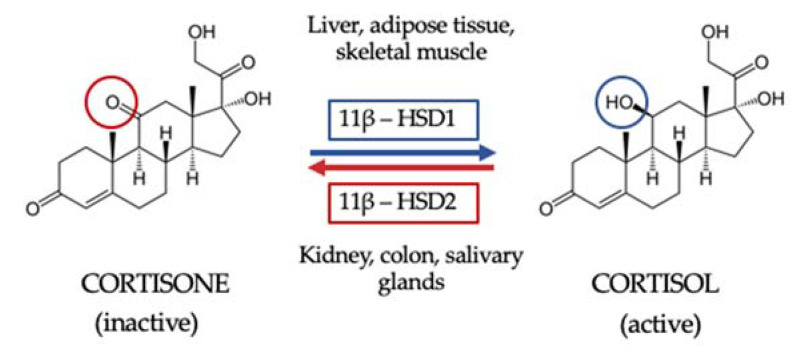
The 11β hydroxysteroid dehydrogenase shuttle. The 11β hydroxysteroid dehydrogenase type 1 isoform (11BHSD1) is expressed in key metabolic tissues such as the liver, adipose tissue and skeletal muscle, and its function is to convert inactive cortisone to active cortisol. The 11BHSD type 2 isoform (11HSD2) is expressed by kidney, colon and salivary glands and catalyzes the inactivation of cortisol to cortisone, thus not only protecting the mineralocorticoid receptor from the occupancy of cortisol but also providing the substrate for 11BHSD1 in peripheral tissue.

**Figure 2 ijms-23-00673-f002:**
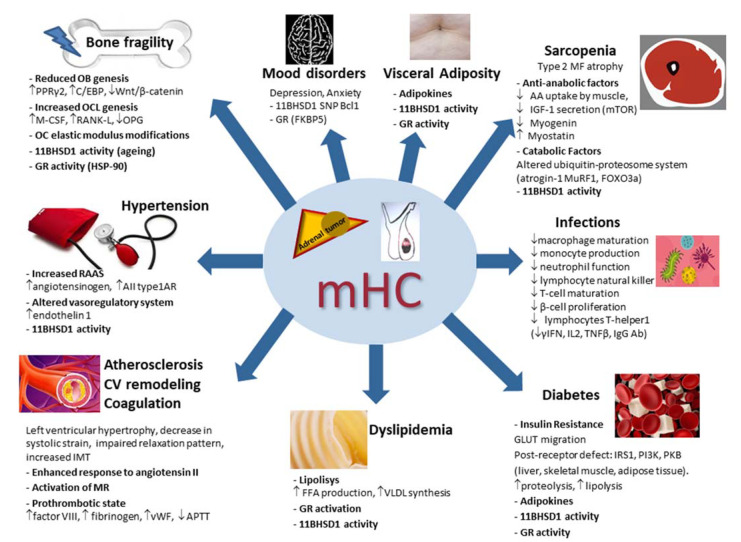
Main pathophysiologic mechanisms of the systemic consequences of mild hypercortisolism. mHC: mild hypercortisolism; OB: osteoblast; OCL: osteoclast; OC: osteocytes; PPRγ2: peroxisome proliferator-activated receptor γ2; C/EBP: CCAAT/enhancer binding protein family; M-CSF: macrophage stimulating factor; RANK-L: receptor activator of nuclear factor kappa B ligand; 11BHSD1: 11β hydroxysteroid dehydrogenase type 1; HSP-90: heat shock protein 90; AIItype1AR: angiotensin II; type 1A receptors; MR: mineralocorticoid receptor; vWF: von Willebrand factor; APTT: activated partial thromboplastin time; IMT: intima–media thickness; CV: cardiovascular; GR: glucocorticoid receptor; VLDL: very-low-density lipoprotein; FFA: free fatty acid; GLUT: glucose transporter; IRS1: insulin receptor substrate-1; PKB: protein kinase B; PI3K: phosphatidylinositol-3 kinase; AA: amino acid; IGF-1: insulin growth factor 1; mTOR: mammalian target of rapamycin (mTOR); MF: muscle fibers; MuRF1: muscle-ring finger protein-1; FOXO3a: forkhead box O3a; SNP: single nucleotide polymorphism; γIFN: γ-interferon; IL2: interleukin 2; TNFβ: tumor necrosis factor β; Ab: antibodies. ↑: increased; ↓: decreased.

**Figure 3 ijms-23-00673-f003:**
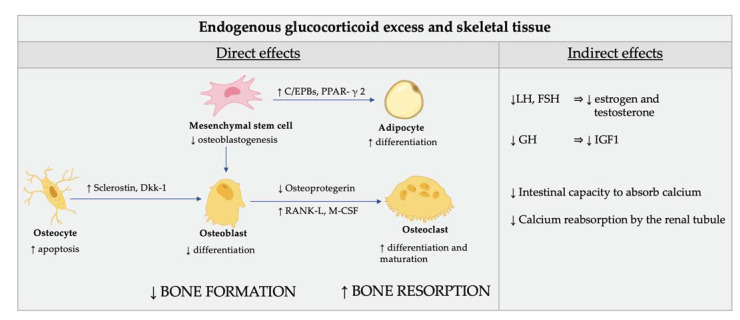
Direct (cellular) and indirect effect of glucocorticoid excess on skeletal tissue. Glucocorticoids (GC) reduce the osteoblastogenesis from the mesenchymal stem cells by directing their differentiation to adipocytes rather than osteoblasts. The mechanisms involved in this redirection of mesenchymal cells include induction of the nuclear factors of the CCAAT/enhancer binding protein family (C/EPBs) and of the peroxisome proliferator-activated receptor γ 2 (PPAR-γ 2). GC also affect the function of osteocytes by modifying the elastic modulus surrounding osteocytic lacunae and induce cell apoptosis. In addition, GC induce the production of sclerostin and Dickkopf-related protein 1 (Dkk-1) by osteocytes which impairs osteoblasts’ function through inhibition of Wnt signaling. The endogenous cortisol excess also induces differentiation and maturation of osteoclasts by increasing the expression of the macrophage colony-stimulating factor (M-CSF) and of the receptor activator of nuclear factor kappa-Β ligand (RANK-L) and decreasing the expression of its soluble decoy receptor, osteoprotegerin, in stromal and osteoblastic cells. As a result, the normal maintenance of the bone is impaired and the bone biomechanical properties are compromised. The indirect effects of GC on bone are mediated by their inhibition of the gonadotropins release, which, in turn, affects estrogen and testosterone production. Finally, GC plays a role in the inhibition of the production of growth hormone and insulin-like growth factor, which are important stimulators of osteoblasts. Calcium homeostasis is also adversely affected by GC. The intestinal capacity to absorb calcium is lowered and calcium reabsorption by the renal tubule is inhibited by GC.

**Figure 4 ijms-23-00673-f004:**
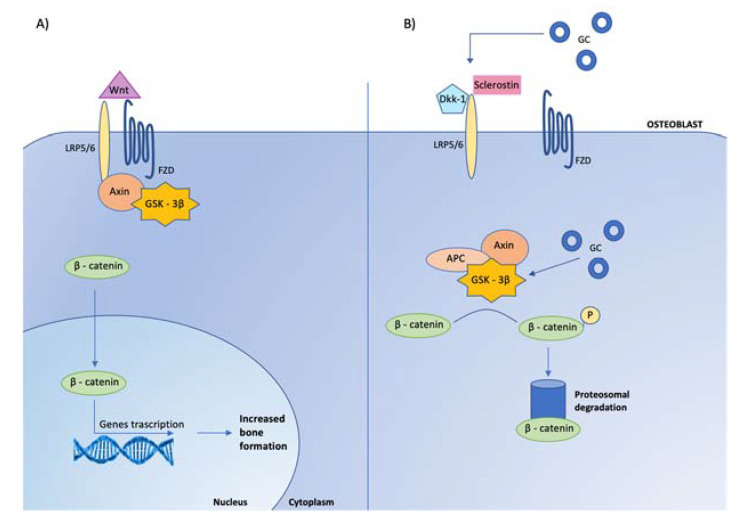
Effects of glucocorticoids on canonical Wnt signaling in osteoblasts. Glucocorticoids (GC) inhibit osteoblast cell differentiation by opposing Wnt/β-catenin signaling. (**A**) In osteoblasts, when Wnt is present, it binds to specific receptors, called frizzled (FZD) to coreceptors, low-density lipoprotein receptor related proteins (LRP-5/6), leading to the inhibition of glycogen-synthase kinase-3β (GSK-3β) activity, via its link with LRP-5/6 mediated by the protein Axin. When GSK-3β is not active, stabilized β-catenin translocates to the nucleus, where it associates with transcription factors to regulate gene expression involved in osteoblast activity. (**B**) When Wnt is absent, the complex GSK-3β-Axin- Adenomatous Polyposis Coli (APC) is activated, leading to β-catenin phosphorylation (P) and consequently to its degradation by ubiquination (proteosomal degradation). The Wnt-β-catenin signaling pathway is regulated by several inhibitors, among which are Dickkopf-related protein 1 (Dkk-1) and sclerostin, that prevent Wnt from binding to its receptor complex. Glucocorticoids enhance the expression of Dickkopf and sclerostin by osteocytes and maintain GSK 3-β in an active state, leading to the inactivation of β-catenin.

**Figure 5 ijms-23-00673-f005:**
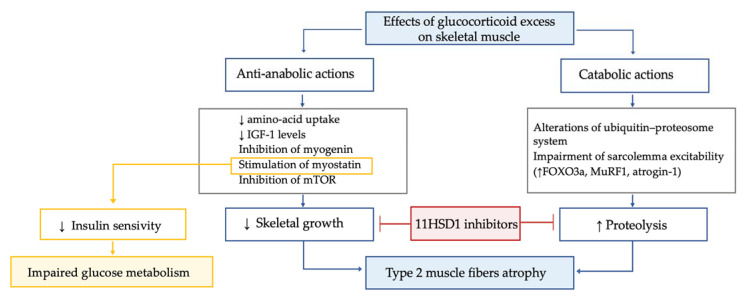
Effects of glucocorticoid excess on skeletal muscle tissue.

**Table 1 ijms-23-00673-t001:** Comparison of characteristics of Cushing’s syndrome and mild hypercortisolism.

	Cushing’s Syndrome	Mild Hypercortisolism
Definition	Large group of signs and symptoms that reflect prolonged and inappropriately high exposure of tissue to glucocorticoids	Alteration of hypothalamic–pituitary–adrenal axis secretion in the absence of signs or symptoms of cortisol excess
Prevalence	1/500,000	0.8–2/1000
Origin	ACTH-dependent: 80%	ACTH-dependent: 10–15%
	ACTH-independent: 20%	ACTH-independent: 85–90%
Specific signs and symptoms	Easy bruising	None
	Facial plethora	
	Proximal myopathy (or proximal muscle weakness)	
	Striae (especially if reddish purple and >1 cm wide)	
Associated conditions	Bone fragility	Bone fragility
	Diabetes	Diabetes
	Hypertension	Hypertension
	Obesity	Obesity
	Dyslipidemia	Dyslipidemia
	Mood disorders	Mood disorders
Diagnostic delay	3–5 years	Not known

**Table 2 ijms-23-00673-t002:** Comparison between macronodular and micronodular bilateral adrenocortical hyperplasia.

	Macronodular Bilateral Adrenocortical Hyperplasia	Micronodular Bilateral Adrenocortical Hyperplasia
Size	Bilateral benign adrenal macronodules >1 cm.	Bilateral adrenal nodules <1 cm in diameter. Primary pigmented nodular adrenal disease is the most frequent form.
Cortisol secretion degree/clinical presentation	Associated with variable degree of cortisol excess. Responsible for less than 2% of all cases of Cushing’s syndrome.	Clinically overt cortisol excess (Cushing’s syndrome).
Age of onset	Late median age in sporadic cases (around 55 years).	Diagnosis is often made before the age of 30 years, with 50% of patients being less than 15 years.
Syndromic cases	Rarely may be part of hereditary familial tumor syndromes including multiple endocrine neoplasia type 1 (*MEN1*), familial adenomatous polyposis (*APC*) and hereditary leiomyomatosis and renal cell cancer syndrome (fumarate hydrogenase, *FH*).	Primary pigmented nodular adrenal disease can be the presenting manifestation of Carney complex (PRKAR1A mutations).
Familial cases	Related to germline mutations of the Armadillo Repeat Containing 5 (ARMC5). A mutation of ARMC5 gene is found in around 20–25% of all bilateral macronodular adrenocortical hyperplasia cases (40% in patients with overt Cushing’s syndrome and 11% in patients with mHC).	Isolated micronodular bilateral adrenocortical hyperplasia can be related to germline PRKA1RA mutation (c.709-7del6 is the most frequent form).
Sporadic cases	Cortisol secretion is in part regulated by the expression of multiple aberrant G protein-coupled receptors (GPCRs). Many of these aberrant receptors stimulate the cAMP/PKA pathway, as does ACTH in normal adrenals.	Micronodular bilateral adrenocortical hyperplasia can be sporadic or familial.

**Table 3 ijms-23-00673-t003:** Genetic alterations in unilateral adrenal adenomas associated with Cushing’s syndrome and mild hypercortisolism (mHC).

	Adrenal Cushing’s Syndrome		Adrenal mHC	
	PRKACA (n. Mutated Patients/ n. Studied Patients)	CTNNB1 (n. Mutated Patients/ n. Studied Patients)	PRKACA (n. Mutated Patients/ n. Studied Patients)	CTNNB1 (n. Mutated Patients/ n. Studied Patients)
Cao et al. [43]	57/87	1/87	-	-
Beuschlein et al. [44]	22/59	0/59	0/40	0/40
Goh et al. [45]	13/36	3/36	3/27	6/27
Di Dalmazi et al. [46]	22/64	-	0/36	-
Sato et al. [47]	33/55	0/55	1/9	0/9
Thiel et al. [48]	11/36	5/36	1/22	8/22
Ronchi et al. [49]	-	7/39	-	19/35
Overall (%)	158/337 (46%)	16/312 (5%)	5/109 (4.6%)	33/133 (25%)

## Data Availability

Not applicable.

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
