# Peer review of "Pathophysiology of Mild Hypercortisolism: From the Bench to the Bedside"

_ijms, 2022, doi:10.3390/ijms23020673_

Round 1

Reviewer 1 Report

This review summarizes clinical, biochemical and genetic characteristics of mild hypercorticalism (mHC), as compared to the classical pituitary/adrenal Cushing sindromes (CSs). The objective is to show that mHC is not simply a mild CS, but it may have an indipendent pathogenesis and, hence, is a separate clinical syndrome.

Authors have collected and discussed a large number of published data, including those from their laboratories (see self-citations). Taken together, these data suggest, although not convincingly, that mHC can be, at least in its genetic substrate, different from CSs.

Despite the present available data are insufficient to adhere completely to the author hypothesis, the review raises interest, evidences still pending questions and stimulates further studies in mHC. Therefore, it can be accepted after minor revision.

I have a some suggestions:

1.-What is the prevalence of acquired mHC vs genetic predisposition?

2.-It is not clear what could be the pathogenetic pathway(s) of the acquired mHC. The stressful conditions in the early life, with permanent changes (what?) in adult life suggest a relevant role of epigenetic mechanisms. There are data on this topic?

3.-Due to the plethora of acronyms some paragraphs are difficult to read.

4.-Ref #4 - should be completed.

Author Response

This review summarizes clinical, biochemical and genetic characteristics of mild hypercorticalism (mHC), as compared to the classical pituitary/adrenal Cushing sindromes (CSs). The objective is to show that mHC is not simply a mild CS, but it may have an indipendent pathogenesis and, hence, is a separate clinical syndrome.

Authors have collected and discussed a large number of published data, including those from their laboratories (see self-citations). Taken together, these data suggest, although not convincingly, that mHC can be, at least in its genetic substrate, different from CSs.

Despite the present available data are insufficient to adhere completely to the author hypothesis, the review raises interest, evidences still pending questions and stimulates further studies in mHC. Therefore, it can be accepted after minor revision.

I have a some suggestions:

1.-What is the prevalence of acquired mHC vs genetic predisposition?

We thank the referee for the suggestion. A potential genetic cause has been suggested in about 40% of patients with mHC (5% PRKARCA mutations, 25% CTNNB1 mutations and 11% ARMC5 mutations). This is now reported (lines 347-352) in the new version of the manuscript.

2.-It is not clear what could be the pathogenetic pathway(s) of the acquired mHC. The stressful conditions in the early life, with permanent changes (what?) in adult life suggest a relevant role of epigenetic mechanisms. There are data on this topic?

We thank the reviewer for the suggestions. In the new version of the manuscript we have discussed the epigenetic mechanisms possibly linked to mHC pathogenesis (lines 234-248, lines 352-360)

3.-Due to the plethora of acronyms some paragraphs are difficult to read.

The acronyms have been reduced as much as possible throughout the manuscript. The acronyms for Cushing’s syndrome (CS), hypothalamus–pituitary–adrenal axis (HPA), single nucleotide polymorphism (SNPs) and knock-out (KO), bone mineral density (BMD), intima-media thickness (IMT), type 2 diabetes (T2DM), lipoprotein receptor related proteins (LRP), glycogen-synthase kinase-3β (GSK-3β), macronodular adrenal hyperplasia (BMAH), bilateral adrenocortical hyperplasia (BAH), forkhead box O3a (FOXO3a) and von Willebrandt factor (vWF) have been removed.

4.-Ref #4 - should be completed.

The ref#4 has been deleted as it was redundant

Reviewer 2 Report

The manuscript is well written and conceived.

Since, the diagnostic issue is only cited in the discussion, please consider to add a brief chapter in the first part

Author Response

Reviewer 2

The manuscript is well written and conceived.

Since, the diagnostic issue is only cited in the discussion, please consider to add a brief chapter in the first part

We agree. Consequently a brief chapter on the diagnosis of mild hypercortisolism has been added in the Introduction section (lines 76-87)

Reviewer 3 Report

The authors in the present paper aimed to resume state of art on the pathophysiology of mild hypercortisolism, differentiating it from overt hypercortisolism and from Cushing disease. The aim is well defined and interesting; however, the current form has some major issues that make it unacceptable for publication in IJMS.

-the manuscript has evident English mistakes and need to be extensively revised by a mother tongue: here you are some examples:

  • Line 29: comma should be removed after excess
  • Line 71: suggested, not suggest
  • Line 72: such as patients with scarcely controlled (please add with
  • Line 72 and 74 : the second scarcely could be changed with another adverb with the same meaning
  • Line 88: may cause to, please remove to
  • Line 182: remove have from have helped
  • Line 196: and those with mHC instead of and mHC
  • Line 202: is related instead of is relates
  • Line 299: “comparison of…” instead of “…..compared”

In addition, the following sentences need to be rewritten because they are not so clear:

  • Line 31-32 (and lines 148-149)
  • Line 41: the different sources ….and their (plural form is better than singular)
  • Line 172-174: not clear
  • Line 176-177: s germline mutation of epigenetic alteration…this sentence has no sense
  • 201-202 not clear the link between increased apoptosis and wnt pathway, please better explain how these roles could be both exerted by ARMC5ù
  • Line 433-434
  • Line 608
  • Line 717-719
  • Line 724-728

-mat and meth section should be added containing research criteria used for the selection of publication considered for the present review (keywords, time considered, type of paper, searching databases included for the search)

Line 286: please define the database listing for the genetic variants (link). Reference 16 is from 2014. Knowledge of variants is constantly updated. Would you please give a more recent reference?

- the paragraph on “Pathophysiology of pituitary mild hypercortisolism” does not give a clear explanation for mHC and ACTH tumors and their difference from ACTH adenomas responsible for CD. Please consider this and try to reformulate the paragraph

- the overall paper is very long and gives a lot of written information. I think that some figures could better help to explain and to resume. Please consider a figure for the following part of the paper:

2.3, line 181-207

2.3 line 217-248

4.1 line 403-428

4.5 611-642

The enzyme 11bhsd1 has here a pivotal role in GC and mHC. A diagram/figure could help to keep in mind the different functions of this enzyme

Author Response

The authors in the present paper aimed to resume state of art on the pathophysiology of mild hypercortisolism, differentiating it from overt hypercortisolism and from Cushing disease. The aim is well defined and interesting; however, the current form has some major issues that make it unacceptable for publication in IJMS.

The manuscript has evident English mistakes and need to be extensively revised by a mother tongue: here you are some examples:

Line 29: comma should be removed after excess

Line 71: suggested, not suggest

Line 72: such as patients with scarcely controlled (please add with

Line 72 and 74 : the second scarcely could be changed with another adverb with the same meaning

Line 88: may cause to, please remove to

Line 182: remove have from have helped

Line 196: and those with mHC instead of and mHC

Line 202: is related instead of is relates

Line 299: “comparison of…” instead of “…..compared”

We apologize. The new version of the manuscript has been revised by a mother tongue and the suggested correction have been made

In addition, the following sentences need to be rewritten because they are not so clear:

Line 31-32 (and lines 148-149)

Line 41: the different sources ….and their (plural form is better than singular)

Line 172-174: not clear

Line 176-177: s germline mutation of epigenetic alteration…this sentence has no sense

201-202 not clear the link between increased apoptosis and wnt pathway, please better explain how these roles could be both exerted by ARMC5

Line 433-434

Line 608

Line 717-719

Line 724-728

In the new version of the above-mentioned sentences have been rewritten as suggested (lines 31-32, lines 183-184, lines 250-254, line 256-257, lines 254-264, lines 537-542, lines 726-729, lines 850-853, lines 857-859)

Mat and meth section should be added containing research criteria used for the selection of publication considered for the present review (keywords, time considered, type of paper, searching databases included for the search)

As this manuscript was not aimed to be a Systematic Review, a Material and Method Section has not been included. However, we agree that such a Section could be useful for the readers and consequently, a Material and Method Section has been added in the new version of the manuscript (lines 105-113)

Line 286: please define the database listing for the genetic variants (link). Reference 16 is from 2014. Knowledge of variants is constantly updated. Would you please give a more recent reference?

A link for the database listing for the genetic variants has been added (lines 90-92) and the reference 16 has been updated)

- the paragraph on “Pathophysiology of pituitary mild hypercortisolism” does not give a clear explanation for mHC and ACTH tumors and their difference from ACTH adenomas responsible for CD. Please consider this and try to reformulate the paragraph

The Paragraph on “Pathophysiology of pituitary mild hypercortisolism” has been extensively modified (lines 195-224, 234-248).

- the overall paper is very long and gives a lot of written information. I think that some figures could better help to explain and to resume. Please consider a figure for the following part of the paper:

2.3, line 181-207:

This part has been revised but, in our opinion is hardly reducible. A table has been added (table 2)

2.3 line 217-248:

This part has been summarized (lines 302-328) and a table has been added (table 3)

4.1 line 403-428:

This part has been summarized (lines 537-548) and 2 figures have been added (figure 3 and 4)

4.5 611-642

This part has been summarized (lines 750-772) and the main mechanisms of GC effects on muscle are now depicted in the figure 5

The enzyme 11bhsd1 has here a pivotal role in GC and mHC. A diagram/figure could help to keep in mind the different functions of this enzyme

In the new version of the manuscript, the figure 1, depicts the main functions of the 11BHSD shuttle

Round 2

Reviewer 3 Report

the second revision of the paper is for me ok, the only change is the GC between comas at line 88, that is a repetition of an abbreviation that could have been added before in the manuscript, if necessary